# Novel Pathogen–Plant Host Interaction: *Colletotrichum jiangxiense* and *Fraxinus americana* L. (White Ash) in a Sentinel Garden in China

**DOI:** 10.3390/plants12234001

**Published:** 2023-11-28

**Authors:** Lin Chang, Yilin Li, Ziwen Gao, Pierluigi (Enrico) Bonello, Michelle Cleary, Isabel A. Munck, Alberto Santini, Hui Sun

**Affiliations:** 1Collaborative Innovation Center of Sustainable Forestry in Southern China, College of Forestry, Nanjing Forestry University, Nanjing 210037, China; linchang@njfu.edu.cn (L.C.); yilinli@njfu.edu.cn (Y.L.); gzw13782577313@163.com (Z.G.); 2Department of Plant Pathology, The Ohio State University, Columbus, OH 43210, USA; bonello.2@osu.edu; 3Southern Swedish Forest Research Centre, Swedish University of Agricultural Sciences, 230 53 Alnarp, Sweden; michelle.cleary@slu.se; 4USDA Forest Service, Durham, NH 03824, USA; isabel.munck@usda.gov; 5Institute for Sustainable Plant Protection—C.N.R., 50019 Florence, Italy; alberto.santini@cnr.it

**Keywords:** sentinel planting garden, *Fraxinus americana* L., leaf spot, *Colletotrichum jiangxiense*, pathogenicity

## Abstract

*Fraxinus americana* L. (white ash), a native North American tree commonly cultivated for its ornamental qualities, displayed symptoms of leaf spot disease in a sentinel garden located in Nanjing, Jiangsu, China, in 2022. This disease led to premature leaf shedding, adversely affecting the plant’s growth and substantially diminishing its ornamental value. Potential fungal pathogens were isolated from the diseased leaves and the subsequent application of Koch’s postulates confirmed the pathogenicity of the fungal isolates (BL-1, BL-2). Through a combination of multi-locus phylogenetic analysis, including ITS, *ACT*, *ApMat*, *CAL*, *CHS-1*, *GAPDH*, and *TUB2*, alongside morphological assessments, the fungus was conclusively identified as *Colletotrichum jiangxiense*. This represents the first record of *C. jiangxiense* affecting white ash, highlighting the important role of sentinel gardens in uncovering novel pathogen–plant host interactions.

## 1. Introduction

*Fraxinus americana* L. (white ash), commonly known as white ash, is a native North American tree species. Its popularity in the gardening industry is attributed to its vibrant autumn colors and rapid growth rate [1]. In recent years, white ash has gained widespread recognition as a landscape tree species in southern and southwestern China [2]. White ash wood has high impact resistance and is often used in the production of boats and baseball bats [3]. Additionally, its calcium-rich leaves are highly favored by earthworms and offer soil enrichment benefits [4,5].

Members of the *Colletotrichum* genus, ranked as the eighth most significant fungi by plant pathologists, exhibit a versatile role as phytopathogens, epiphytes, humic inhabitants, or endophytes [6]. As pathogens, *Colletotrichum* spp. exhibit a broad host range, posing threats not only to plants but also to humans, causing subcutaneous infections and keratitis [7,8]. They are also known to infect insects [9] and are responsible for disease in various herbaceous and woody plants, particularly impacting food crops, fruits, vegetables, and ornamental plants, resulting in substantial yield or economic losses [10,11,12,13]. Moreover, reports of multiple *Colletotrichum* species co-infecting hosts are also increasing [14,15]. The diversity of the living environments of *Colletotrichum* species and their destructive capabilities highlight the urgent need for scientific research in this area.

A sentinel garden is an effective means for studying and providing an early warning about invasive or alien pests [16,17]. The method involves growing and monitoring plant species in areas outside their native range, identifying risks posed by insects and pathogens, thereby providing valuable early warning information to the country of origin [18]. For example, Roques et al. established a sentinel nursery in China to plant five ornamental tree species from Europe to help identify the potential risks from trade-introduced insects [19]. A sentinel garden is also a useful tool for detecting new host associations between pests and diseases [20]. For instance, during 2012–2013, Kenis et al. conducted sentinel surveillance in China on various Asian ornamental tree species exported to Europe. Ultimately, they discovered 105 insect–host associations on sentinel plants, 90% of which were recorded for the first time; they also found five pathogens associated with trees, each causing different symptoms [21,22]. According to Botanic Gardens Conservation International (BGCI), more and more countries are participating in sentinel research [23]. Global mutual support and coordination can fully leverage the value of sentinel gardens for plant health.

In 2021, while conducting a disease survey on sentinel plants as part of a sentinel garden project, characteristic leaf spot symptoms were observed on white ash saplings (5 years old). This study aims to determine the causal agent of leaf spot disease in white ash through morphological observation, molecular identification, and pathogenicity tests.

## 2. Results

### 2.1. Field Symptoms and Fungal Isolation

In the three surveys, white ash exhibited an average incidence of 40.7%, with a disease severity rating of 1. The initial symptoms appeared as small dark brown lesions on the leaf surface, which progressively enlarged and clustered into large irregular necrotic spots. The necrotic areas caused leaf curl and eventual defoliation, leading to diminished plant vigor (Figure 1A,B). A total of 53 fungal isolates were obtained from the diseased leaves collected during the surveys. Based on colony morphology, these isolates were categorized into three types, representing *Colletotrichum*, *Alternaria*, and *Diaporthe*. The detailed isolation frequencies for each isolate type in the three surveys are presented in Table 1.

### 2.2. Pathogenicity Test

In vitro, five days post-inoculation with fungal hyphae, only *Colletotrichum* sp. caused brown spot symptoms on detached leaves, while the control leaves showed no symptoms (Figure 1C,D). Subsequently, the spore suspension of the *Colletotrichum* sp. selected was inoculated in the leaves attached to the saplings and, after 7 days, the inoculated leaves displayed brown spots resembling early field symptoms, while the control leaves remained healthy (Figure 1E,F). The same fungus was successfully re-isolated from the lesions, with no other fungi present on the control leaves. These results fulfilled Koch’s postulates, confirming that the *Colletotrichum* sp. isolates were the causal agents of the leaf spot on white ash.

### 2.3. Morphological Identification of the Pathogen

*Colletotrichum* sp. exhibited white colonies with aerial mycelium on PDA (Figure 1G). The conidia were aseptate, hyaline, smooth walled, conical, or subcylindrical, occasionally with a round apex and a slightly pointed base, slightly constricted in the middle, measuring 9.2–17.5 × 3.5–6.8 µm, with a mean ± standard deviation (SD) of 13.2 ± 1.2 × 5.2 ± 0.2 µm (Figure 1H). The appressoria were black, solitary, smooth, nearly spherical, or ellipsoidal, measuring 14.4–17.5 × 8.4–14.3 µm (Figure 1I). The representative strains, BL-1 and BL-2, in *Colletotrichum* sp. were selected for subsequent molecular identification.

### 2.4. Molecular Identification

The BLAST analysis showed that the sequences of ITS, *ACT*, *CAL*, *TUB2*, *CHS-1*, *ApMat*, and *GAPDH* of the BL-1 and BL-2 isolates were highly matched (>99%) to those of *Colletotrichum jiangxiense*. The sequence accession numbers for isolates BL-1 and BL-2 are provided in Table 2. The cladistic clustering results from the seven-locus phylogenetic tree, constructed using the maximum likelihood method, concurred with the BLAST analysis (Figure 2). Based on the morphological and phylogenetic analysis, the pathogen causing white ash leaf spot in China was identified as *C. jiangxiense*.

## 3. Discussion

Owing to the diversity within the *Colletotrichum* species, relying solely on ITS and the morphological characteristics may not be enough for precise species-level classification. Thus, the integration of multi-gene analysis is essential for the accurate identification of the *Colletotrichum* species [24]. Various genetic loci, especially within species complexes, have been employed to obtain comprehensive molecular information for species differentiation [25]. Additional loci, including *ACT*, *CAL*, *CHS-1*, *GAPDH*, and *TUB2* genes, have been used for the *Colletotrichum* species complex [26,27]. In this study, we utilized a multi-locus phylogenetic approach based on those sequences mentioned above, coupled with the morphological characteristics. This approach conclusively identified the pathogen as *C. jiangxiense*.

The *Colletotrichum* genus contains approximately 600 species and attacks more than 3200 monocot and dicot species [28]. Phytopathogens within this genus can not only survive on the plants they infect, but can also form a mutualistic or symbiotic relationship with other plants [29]. *C. jiangxiense* was first recorded and described in 2015 as an endophyte within *Camellia* spp. [30]. However, in recent years, *C. jiangxiense*, has emerged as a pathogen affecting a variety of fruits and ornamental plants in China, significantly impacting host plants [31,32,33]. It has also been identified as the cause of avocado anthracnose in Mexico [34]. Currently, there are no reports on plant diseases caused by *C. Jiangxiense* in North America, the native habitat of white ash. Nonetheless, East Asia, particularly China, is one of the regions with the most substantial trade in live plants imported from and exported to North America [35]. In favorable conditions, there is a possibility of a transition from commensal or low-pathogenic behavior to highly pathogenic behavior [36,37], potentially resulting in the pathogenic behavior of *C. jiangxiense* affecting local white ash tree species and other plants in North America.

To the best of our knowledge, this study represents the first report on anthracnose disease caused by *C. jiangxiense* in white ash in China. This finding lays the foundation for developing sustainable and effective management strategies for combating this disease. Our study is expected to contribute to future management programs on this reported pathogen–host interaction in China.

## 4. Materials and Methods

### 4.1. Sampling and Fungal Isolation

In 2022, regular disease surveys were conducted at three sampling intervals (May, July, and September) in the sentinel garden, located in Lishui district, Jiangsu Province, China. During these surveys, leaf spots were observed on five-year-old white ash saplings. The disease incidence and severity were assessed on a scale ranging from 0 to 6, following the sentinel study survey protocol described by Morales-Rodríguez et al. [38]. For each survey, ten leaves exhibiting evident symptoms were collected and subjected to fungal isolation. Small (5 × 5 mm) tissue samples were excised from the necrotic tissue margins, resulting in a total of 20 pieces. These tissues were surface sterilized using established procedures [39], subsequently placed onto potato dextrose agar (PDA) plates, and incubated at 25 °C in darkness for 3 days. Pure cultures were obtained by transferring mycelial edges onto fresh PDA plates. The preliminary classification of the isolates was performed based on morphology.

### 4.2. Pathogenicity Test

In order to determine the pathogenicity of the isolates, inoculation tests were conducted on detached and attached leaves from white ash. Detached healthy field-collected leaves were washed for 15 min under tap water, dried, and wounded with sterile needles. Two representative fungal isolates were selected from each type of isolated fungi. An agar plug (6 mm in diameter) pre-colonized by these representative isolates was gently placed onto the wound and removed after 24 h. An agar plug without fungal pre-colonization served as a control. Three leaves were inoculated for each representative isolate. Following inoculation, the leaves were placed in a Petri dish to maintain humidity and cultured in a 25 °C incubator. Subsequently, initially identified pathogenic strains from the detached leaf test were inoculated on sapling leaves for further pathogenicity determination using conidial suspension. Healthy leaves were wounded with a sterile needle, and 10 µL conidial suspensions (106 conidia/mL) of the isolates were inoculated. Three leaves were inoculated for each isolate, while healthy leaves treated with sterilized H_2_O water were used as a control group. All inoculated leaves were covered with sealed bags, and sterilized water was sprayed into the bags daily to maintain humidity. The inoculated saplings were kept in a room with a constant temperature of 25 ± 1 °C.

### 4.3. Morphological Identification of the Pathogens

The leaf pathogenic isolates were cultured on PDA at 25 °C in darkness. After a 5-day incubation period, the colony characteristics, including the colony color, texture, conidia morphology, and appressoria, were recorded. For accurate morphological descriptions and size measurements for both the conidia and appressoria, a Zeiss Axio Imager A2m microscope was employed for observation (*n* = 30). The appressoria were induced from the conidia using a slide culture technique, as described by Cai et al. [40].

### 4.4. DNA Extraction and PCR Amplification

Fungal genomic DNA was extracted from the aerial hyphae of the representative cultures grown for 5 days using the cetyltrimethylammonium bromide (CTAB) extraction procedure [41]. The polymerase chain reaction (PCR) was employed to amplify the internal transcribed spacer (ITS) region and gene loci, namely *ACT*, *ApMat*, *CAL*, *CHS-1*, *GAPDH*, and *TUB2*, using specific primers ITS1/ITS4 [42], ACT-512F/ACT-783R [43], AM-F/AM-R [44], CL1C/CL2C [45], CHS-79F/CHS-345R [43], GDF/GDR [46], and Bt2a/Bt2b [47], respectively. The PCR reaction conditions and sequences of the primers are detailed in Table 2. The PCR products were purified and sequenced by Nanjing Sipujin Biotechnology Co., Ltd. (Nanjing, China). The DNA sequences for each region/gene obtained were submitted to the GenBank at the National Center for Biotechnology Information (NCBI), and the accession numbers are detailed in Table 2.

### 4.5. Phylogenetic Analysis

The obtained sequences of ITS, *ACT*, *ApMat*, *CAL*, *CHS-1*, *GAPDH*, and *TUB2* were subjected to a BLAST comparison against the GenBank database. The DNA sequences from the type specimens of the species and phylogenetically closely related species were selected for phylogenetic analysis [30]. Gene sequences alignment was performed manually using MAFFT (ver. 7.313) and BioEdit (ver. 7.0.9.0) [48,49]. Individual single-gene sequences were concatenate using the PhyloSuite (ver. 1.2.1) to generate composite sequences. A maximum likelihood (ML) phylogenetic tree was constructed, incorporating five genetic regions (ITS, *ACT*, *CAL*, *TUB2*, *CHS-1*, *ApMat*, and *GAPDH*). The branch stability was assessed through 1000 bootstrap replications, and the phylogenetic trees were visualized using FigTree (ver. 1.4.4).

## 5. Conclusions

This study highlights the importance of accurate and rapid pathogen identification for disease management strategies. Sentinel garden research can serve as a sensitive disease surveillance tool and method.

## Figures and Tables

**Figure 1 plants-12-04001-f001:**
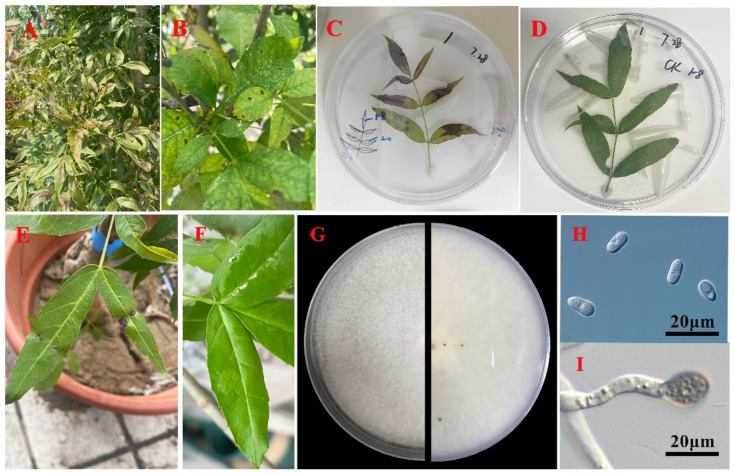
Leaf spot symptoms on white ash and morphological characteristics of *Colletotrichum* sp. (**A**,**B**) Initial and subsequent field symptoms observed on white ash leaves; (**C**,**D**) symptoms exhibited on detached leaves 5 days post-inoculation with a mycelium of *Colletotrichum* sp. and a control (*n* = 3); (**E**,**F**) symptoms exhibited on leaves from a sapling 7 days post-inoculation with conidial suspensions of *Colletotrichum* sp. (*n* = 3); (**G**) morphology of the *Colletotrichum* sp. colony’s front side (**left**) and back side (**right**) on a PDA medium; (**H**) conidia; (**I**) appressorium.

**Figure 2 plants-12-04001-f002:**
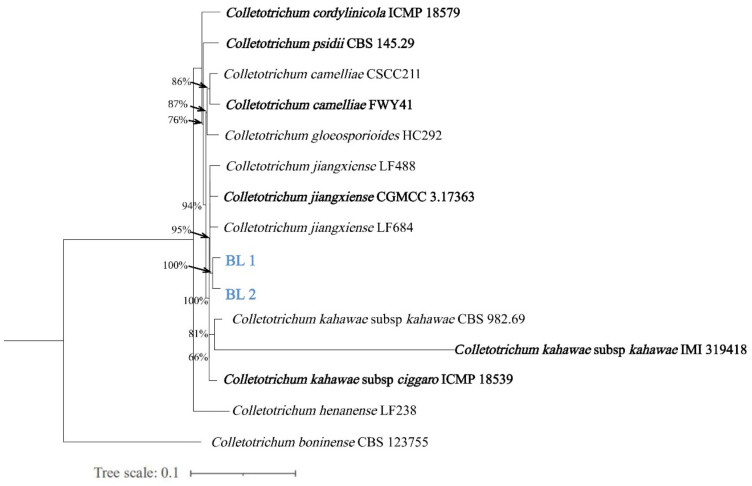
Maximum likelihood tree for the *Colletotrichum* species, constructed using the concatenated dataset (ITS, *ACT*, *CHS-1*, *GAPDH*, *TUB2*, *ApMat*, and *CAL*). The isolates BL-1 and BL-2 obtained from this study formed a monophyletic clade with other isolates from the same species. *C. boninense* was used as the outgroup. The numbers on the branches represent bootstrap values obtained from 1000 bootstrap replications. The ex-type strains are in bold.

**Table 1 plants-12-04001-t001:** Fungi isolates from diseased *Fraxinus americana* leaves across three surveys.

Survey Time	Incidence	Disease Severity (0–6)	Number of Tissues	Number of Colonies		
				*Alternaria* sp.	*Colletotrichum* sp.	*Diaporthe* sp.
May	33% (6)	1	20	9 (45%)	8 (40%)	3 (15%)
July	33% (6)	1	20	9 (45%)	7 (35%)	4 (20%)
September	56% (10)	2	20	8 (40%)	12 (60%)	0

**Table 2 plants-12-04001-t002:** PCR amplification conditions of the DNA for isolate BL-1 and BL-2 and the accession numbers of the gene sequence.

Gene	PCR Primers (Forward/Reverse)	PCR Thermal Cycles (Annealing Temp. in Bold)	Accession Numbers of Representative Isolates	
			BL-1	BL-2
ITS	ITS1/ITS4	94 °C: 3 min, (94 °C: 30 s, **55 °C**: 30 s, 72 °C: 45 s) × 33 cycles, 72 °C: 10 min	OR633454	OR633455
*ACT*	ACT-512F/ACT-783R	94 °C: 3 min, (94 °C: 30 s, **58 °C**: 30 s, 72 °C: 45 s) × 35 cycles, 72 °C: 10 min	OR640125	OR640130
*ApMat*	AM-F/AM-R	94 °C: 3 min, (94 °C: 1 min, 55 °C: 30 s, 72 °C: 1 min) × 35 cycles, 72 °C: 10 min	OR640126	OR640131
*CAL*	CL-1C/CL-2C	95 °C: 3 min, (95 °C: 30 s, **55 °C**: 30 s, 72 °C: 30 s) × 35 cycles, 72 °C: 10 min	OR640127	OR640132
*CHS-1*	CHS-79F/CHS-354R	94 °C: 3 min, (94 °C: 30 s, **58 °C**: 30 s, 72 °C: 45 s) × 35 cycles, 72 °C: 10 min	OR640128	OR640133
*GAPDH*	GD-F1/GD-R1	94 °C: 3 min, (94 °C: 30 s, **58 °C**: 30 s, 72 °C: 45 s) × 35 cycles, 72 °C: 10 min	OR640129	OR640134
*TUB2*	BT-2a/Bt-2b	95 °C: 3 min, (95 °C: 30 s, **55 °C**: 30 s, 72 °C: 30 s) × 35 cycles, 72 °C: 10 min	OR640145	OR640146

## Data Availability

Data are contained within the article.

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
