# Peer review of "Novel Pathogen–Plant Host Interaction: Colletotrichum jiangxiense and Fraxinus americana L. (White Ash) in a Sentinel Garden in China"

_plants, 2023, doi:10.3390/plants12234001_

Round 1
Reviewer 1 Report
Comments and Suggestions for Authors
The authors identified sequences of ITS, ACT, CHS-1, GAPDH, TUB2, ApMat and CAL from Colletotrichum boninense, why is there only one phylogenetic tree in the manuscript? Molecular identification of fungi should be based on multiple sequence analysis to determine whether they are a new species.
More morphological evidence will be needed to show that Colletotrichum boninense is a new species.
The inoculation of pathogenic bacteria has only been done in vitro. It will be better to observe the phenotype if seedlings are inoculated.
How many biological duplications in pathogenicity test were conducted? The authous should shoe it in the caption of Fig 1 and also in the Method.
Comments on the Quality of English LanguageEnglish writing and font format will be needed to be modified, such as H2O in Line 168.
Colletotrichum is sometimes abbreviated, sometimes full name, please unify
Author Response
Dear Reviewer,
First of all, we would like to express our gratitude for reviewing our manuscript and having such valuable comments. We have revised the manuscript thoroughly according to yourcomments and suggestions. Please find the detailed responses below.
Best regards,
Hui Sun

Reviewer 2 Report
Comments and Suggestions for Authors
Although you demonstrate the in vitro pathogenicity of Chinese Colletotrichum isolates from White ash on this host, the experimental design on the molecular identification of these isolates is poorly conceived. Comparing DNA sequences from type specimens is essential for accurate molecular identification of fungal species. Instead, you merely searched DNA sequences of available fungal isolates in a public database where many of these sequences are incorrectly identified. On this basis, therefore, I believe that without comparison of the Colletotrichum isolates from White ash with the type species of C. jiangxiense (culture ex-type CGMCC 3.17363 = LC3463 = LF687), as well as with the phylogenetically closely species, including C. kahawae subsp. kahawae and C. kahawae subsp. ciggaro, this communication cannot be published.
In addition, It recommended attach in the manuscript the DNA sequences of analysed genes not yet available in Genbank. Finally, cultures of fungal isolates should be deposited at recognised institutions.
Author Response

(The authors gave the same response as above.)

Round 2
Reviewer 1 Report
Comments and Suggestions for Authors
Authors has basically given the right responses to my commnets. However, authors show that it is a new type of compatible interaction between Colletotrichum boninense and Fraxinus americana, it is necessary to observe the growth of the pathogen spores/hypha after inoculation. Incompatible interactions also cause local necrosis after pathogen infection.
Comments on the Quality of English LanguageA small amount of English grammar in the article needs to be improved.
Author Response
Point by point response to viewers’ comments
Reviewer 1
(1). Authors has basically given the right responses to my comments. However, authors show that it is a new type of compatible interaction between Colletotrichum boninense and Fraxinus americana, it is necessary to observe the growth of the pathogen spores/hypha after inoculation. Incompatible interactions also cause local necrosis after pathogen infection.
Authors’ response: Authors' Response: Thank you for your comment. In the pathogenicity test, lesions appeared and gradually expanded on the healthy leaves of the seedlings after inoculation with the spore suspension of the C. jiangxienseisolate. Additionally, we successfully reisolated the C. jiangxiense isolate from the diseased leaves. This outcome is attributed to the specific affinity interaction between the pathogen and the host, confirming the Koch’s postulate that C. jiangxiense isolate was the causal agent of the disease.
Reviewer 2 Report
Comments and Suggestions for Authors
Dear Authors,
the proper isolates of Colletotrichum that you have added in the phylogenetic analysis of your study make the molecular identification of C. jiangxiense much more reliable.
Concerning the deposit of isolates at recognised institutions, it is very useful not only in the case of a new fungal species but also for isolates obtained from new hosts, as well as from new geographical areas. In this way, cultures can be preserved over time for the benefit of the scientific community, which can use them to conduct studies on the taxonomy or genetic structure of fungal populations. In fact, by now the most authoritative scientific journals specialising in new reports of plant diseases require the mandatory deposit of fungal isolates with recognised institutions.
Notwithstanding that, I believe that your study can now be published, but I strongly suggest that you proceed to deposit at least one of the C. jiangxiense isolates obtained from white ash in China.
Here are a few suggestions.
Line 100: After “BL-2” add “isolates”, and before “Colletotrichum” add “those of”.
Line 145: Delete the word “characteristic” otherwise specify the shape, average size and number of spots.
Lines 151 and 161: Replace “on” with “onto”
Table 2: “ITS” and “ITS1/ITS4” not in bold
In the conclusion you could add a sentence to emphasise that you report C. jiangxiense in China pathogenic on white ash.
Best regards
Author Response
Point by point response to viewers’ commentsReviewer 2
(1). The proper isolates of Colletotrichum that you have added in the phylogenetic analysis of your study make the molecular identification of C. jiangxiense much more reliable.
Concerning the deposit of isolates at recognised institutions, it is very useful not only in the case of a new fungal species but also for isolates obtained from new hosts, as well as from new geographical areas. In this way, cultures can be preserved over time for the benefit of the scientific community, which can use them to conduct studies on the taxonomy or genetic structure of fungal populations. In fact, by now the most authoritative scientific journals specialising in new reports of plant diseases require the mandatory deposit of fungal isolates with recognised institutions.
Notwithstanding that, I believe that your study can now be published, but I strongly suggest that you proceed to deposit at least one of the C. jiangxiense isolates obtained from white ash in China.
Authors’ response:
Thank you for your comment regarding the deposit of isolates at recognized institutions. We fully agree with the importance of such deposits. We have taken your suggestion and have already submitted two of the C. jiangxiense isolates obtained from white ash in China to a recognized institution for fungi preservation at China Forestry Culture Collection Center (CFCC, http://www.caf.ac.cn/info/1265/37798.htm). The accession number for the isolates may take one or two weeks.
#Line 100: After “BL-2” add “isolates”, and before “Colletotrichum” add “those of”.
Authors’ response: We appreciate your suggestion. These details have been added.
#Line 145: Delete the word “characteristic” otherwise specify the shape, average size and number of spots.
Authors’ response: We appreciate your suggestion. We have removed the word "characteristic".
#Lines 151 and 161: Replace “on” with “onto”
Authors’ response: We appreciate your suggestion. We replace "on" with "onto" in both positions.
#Table 2: “ITS” and “ITS1/ITS4” not in bold
Authors’ response: We appreciate your suggestion. We have corrected these details.
#In the conclusion you could add a sentence to emphasis that you report C. jiangxiense in China pathogenic on white ash.
Authors’ response: We appreciate your suggestion. We have added such emphasis in section 2.5.